# Data-driven study of the COVID-19 pandemic via age-structured modelling and prediction of the health system failure in Brazil amid diverse intervention strategies

**Askery Canabarro** [1,2]☯ *, **Elayne Tenório** [3]☯, **Renato Martins** [4]☯, **Laís Martins** [5]☯, **Samuraí Brito** [1]☯, **Rafael Chaves** [1,6]☯

**1** International Institute of Physics, Federal University of Rio Grande do Norte, Natal, RN, Brazil, **2** Grupo de Física da Matéria Condensada, Núcleo de Ciências Exatas—NCEx, Campus Arapiraca, Universidade Federal de Alagoas, Arapiraca, AL, Brazil, **3** Hospital Santa Casa de Misericórdia de Maceió, Maceió, AL, Brazil, **4** HIV/AIDS Testing and Counseling Center, Itaberaba, BA, Brazil, **5** Superior School of Health Science, Brasília, DF, Brazil, **6** School of Science and Technology, Federal University of Rio Grande do Norte, Natal, Brazil

☯ These authors contributed equally to this work.
* askery.canabarro@arapiraca.ufal.br

**Data Availability Statement:** All data files are available from the Johns Hopkins University COVID-19 database. John Hopkings University

## Abstract

In this work we propose a data-driven age-structured census-based SIRD-like epidemiological model capable of forecasting the spread of COVID-19 in Brazil. We model the current scenario of closed schools and universities, social distancing of people above sixty years old and voluntary home quarantine to show that it is still not enough to protect the health system by explicitly computing the demand for hospital intensive care units. We also show that an urgent intense quarantine might be the only solution to avoid the collapse of the health system and, consequently, to minimize the quantity of deaths. On the other hand, we demonstrate that the relaxation of the already imposed control measures in the next days would be catastrophic.

## Introduction

Coronavirus disease-2019 (COVID-19) caused by the severe acute respiratory syndrome coronavirus (Sars-CoV-2) is a main threat for the public health systems throughout the globe [1–13]. As of April 03, 2020, the world had more than one million (1.016.401) confirmed cases, 53.160 deaths and 211.775 recovered persons [7] since the first suspected case of (COVID-19) on December, 2019, in Wuhan, Hubei Province, China [8], with a frightening propagation speed. So far, 52 countries reported more than 1000 cases from all continents, Antarctica being the only exception with no cases. In Brazil, at the time of writing, the COVID-19 statistics was: 8.066 confirmed cases, 327 deaths and 127 recovered individuals [7].

Therefore, a critical aspect of the COVID-19 pandemic, rather than the percentage of infection and even the mortality rate, is the healthcare system capacity. In Brazil, the number of

https://coronavirus.jhu.edu/map.html;" From there, the person has to click on "Brazil".

**Funding:** The author(s) received no specific funding for this work.

**Competing interests:** The authors have declared that no competing interests exist.

intensive care units (ICUs) up to February 2020 was 36, 939 adult and pediatric beds according to the Cadastro Nacional de Estabelecimentos de Saúde (CNES) available in the DataSUS portal in Ref. [14] (in Portuguese, please see instructions to retrieve the data), with a historical occupancy of not less than 85%, which yields in about 5500 free ICUs [15]. In fact, the report by the Health's National Agency set an ideal occupation target of 80% to 85%, therefore our estimation can be in fact an upper bound. The average number of ICUs per 100.000 inhabitants is around a ten [16], US leading the world with a ratio of 34.7 [17], much below what is expected to be needed as the number of infections approaches its peak. Therefore, no country is really prepared for a devastating amount of critical patients. To cope with that and in order to protect the healthcare systems, in the absence of any efficient medicament and/or vaccine to pharmaceutically detain the fast spread of the disease, many public policies and governmental strategies, termed as non-phamaceutical interventions (NPIs), are been tested amid the epidemic/pandemic situation. Currently, many of such public health measures have been discussed/proposed to decrease the spread of the COVID-19 pandemic, by reducing the social contact in the population and, consequently, the transmission rate of the virus, alleviating the health system and providing time for auxiliary measures (expansion of the system, military hospitals and so on).

Mathematical modelling is a recognized powerful tool to investigate transmission and epidemic dynamics [9–11, 18–24]. Here, we present a data-driven and census-based age-structured mathematical epidemiological model capable of asserting the potential output of many NPI over the Brazillian health system by explicitly computing the basic reproduction rate $R_0$, the evolution of the number of infections and the required quantity of ICUs needed over time. The model is a system of ordinary differential equations (ODEs) for each layer of the age stratification reported for the COVID-19 within the Brazilian population. It is a variation of the SIRD model stratified by age group in which people flow among four states: susceptible (S), infected (I), recovered (R) and dead (D), assuming that the recovered population do not become susceptible, as is suggested for COVID-19. By modeling the current Brazilian scenario, we investigate the effects of applying one of the following NPI policies (or a combination of those): 0) a complete absence of control measures (No NPI); 1) closure of schools and universities (CSU); 2) Social distancing of those over sixty years old (SD60+); 3) voluntary home quarantine (VHQ) and social distance of the entire population as an intense quarantine (IQ).

It is worth mentioning that all of these measures have direct social-economical and ethical implications because it severely reduces individual freedoms, spontaneous social aggregations, interferes in the outflow of industrial products and commodities and so on. However, we deal with a epidemiological model, not capable of inferring the by-product of the NPIs over the overall well-being of the population. Therefore, it is virtually impossible for governments simultaneously to minimize the social-economical impact of COVID-19 pandemic and protect the health system, which means minimising deaths. The case of China indicates in practice that some NPIs (quaratine, social distancing, isolation of infected individuals) can contain the virus spread [6], but as soon as the measures are relaxed another outbreak can take place [25] possibly triggered by imported cases, meaning that, probably, NPIs are going to be necessary from time to time. As we will see, our model predicts that for the Brazilian scenario, only intense quarantine (essentially a combination of NPIs) can prevent the collapse of the health system and consequently save a larger number of lives.

## 1 Model

In the absence of a vaccine, the behavior of the individuals takes a crucial role in the course of an epidemic such as the COVID-19 in Brazil. In a nutshell, and as schematically shown in

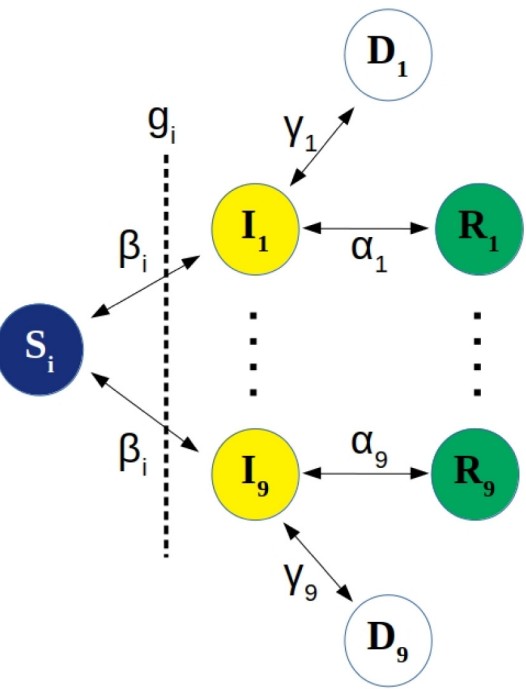

**Fig 1. Schematic representation of the SIRD model under a NPI policy $g_i$.** The individuals can flow among four states in each of the age group $i$: susceptible ($S_i$) to infected ($I_i$) with rate $\beta_i$, who can recover ($R_i$) at a rate $\alpha_i$ or die ($D_i$) at a rate $\gamma_i$, the recovered populations do not become susceptible, as is suggested for COVID-19 [12]. The NPIs policies serve as a barrier to contain the infection. The quantity ($g_i S_i$) represents the fraction of people not respecting the NPI policy, therefore still susceptible.

Fig 1, susceptible (S) individuals can become infected mainly by direct interaction with infected (I) persons at a rate $\beta$, who can recover (R) at a rate $\alpha$ and become resistant to the virus or die (D) at a rate $\gamma$. If, on average, one infected person contaminates one or more individuals, the epidemic is sustainable, otherwise it dies out. This is the definition of the basic reproduction number ($R_0$), which represents the average number of secondary cases from a single infectious case in a totally susceptible population (in the very beginning of the infection the susceptible population corresponds approximately to the total population $N$, $S \approx N$), defined as $R_0 = \beta/(\alpha + \gamma)$. Therefore, it is very important to reduce transmission rate in order to reduce the subsequent reproduction ratio $R_n$ (not to mistaken with the number of recovered individuals) so that $R_n < 1$ and epidemic fades out. In the unlikely worst case scenario in which the government does not impose any measures for containing the infectious disease, $R$ will decrease below unit due to exhaustion of susceptible individuals, at the expense of a tremendous amount of infected and dead individuals, collapsing not only the health system but the economy as well.

Let us define $S(t)$, $I(t)$, $R(t)$, $D(t)$, the number of susceptible, infected, recovered and dead individuals, respectively, at time $t$ in a population of size $N$. The latest Census-based data for Brazil $N = 211, 319, 631$ [26]. In fact, the variables are fractions of the respective compartments, i. e., $s' = S/N$, $i' = I/N$, $r' = R/N$ and $d' = D/N$. Our SIRD model is composed of 36 coupled ordinary differential equations since $i$ belongs to one of the nine age groups as seen in

Table 1, yielding the kinematics of the variables, as follows:

$$\frac{dS_i(t)}{dt} = -\frac{\beta_i}{N}(g_iS_i)I, \tag{1}$$

$$\frac{dI_i(t)}{dt} = \frac{\beta_i}{N}(g_iS_i)I - \alpha_iI_i - \gamma_iI_i, \tag{2}$$

$$\frac{dR_i(t)}{dt} = \alpha_iI_i, \tag{3}$$

$$\frac{dD_i(t)}{dt} = \gamma_iI_i, \tag{4}$$

where $dx/dt$ is the derivative of variable $x$ with respect to time, hence a time variation. In this manner, the corresponding right hand side of each equation provides the variation amount of the variable in time $t$, increasing or decreasing if the amount if positive or negative, respectively. Here, $S_i$ represents the susceptible individuals for age group $i$, $I_i$ stands for infected individuals in age group $i$, $R$ indicates the recovered individuals no more susceptible to infection, and $D$ is the death total. The variable $I = \sum_{i=1}^{9} I_i$ is the total number of infected people in a given time $t$. Here, $\beta_i$, $\alpha_i$ and $\gamma_i$ denote the apparent per day infection (as suggested by data), recovery and mortality rates for the age group $i$, respectively. Note that these parameters do not correspond to the actual per day infection, recovery and mortality rates as the new cases of recovered and deaths come from infected cases several days back in time. However, one can attempt to provide some coarse estimations of the apparent values of these epidemiological parameters based on the reported confirmed cases using an assumption and approach described in the next section, see Table 1. The parameter $g_i$ represents the governmental strategies and/or public policies for the age group $i$ or a combination of them (see Table 2).

In our SIRD model it is assumed that the total number of the population remains constant (dead also included), once the sum Eq 1 implies a null derivative. If the rate at which infected persons need intensive care units (ICU) is known, we can described it as

$$\frac{dH}{dt} = \sum_{i=1}^{9} c_iI_i, \tag{5}$$

**Table 1. Distribution of the population and percentage of the corresponding population currently attending school or university.** As the last information is hard to track, we had to make averages from the data in sources [36, 37] and/or reasonable estimates as for the case of attendance in universities, where it is known that ≈12% [37] of the population from 20–60 years old is attending university, so we distributed almost uniformly among the age groups. Here, 0 means negligible percentage.

| Age | Group (i) | Population[1001] [35] | Sch. or Univ.[1001] [36, 37] |
|---|---|---|---|
| 0 to 9 | 1 | 13.8% | ≈75% |
| 10 to 19 | 2 | 15.0% | ≈60% |
| 20 to 29 | 3 | 16.1% | ≈4% |
| 30 to 39 | 4 | 16.3% | ≈4% |
| 40 to 49 | 5 | 13.7% | ≈2% |
| 50 to 59 | 6 | 11.3% | ≈2% |
| 60 to 69 | 7 | 7.6% | 0 |
| 70 to 79 | 8 | 4.0% | 0 |
| 80 + | 9 | 2.2% | 0 |

**Table 2. Diverse non-pharmaceutical interventions modelled by the parameter $g_i$ ranging from the absence of any control measures (No NPI), closure of schools and universities (CSU), social distancing of people with more than 60 years old (SD60+), voluntary home quarantine (VHQ) and intense quarantine (IQ).** It reflects the fraction of individuals still susceptible after the measure is applied per age group. However, none of them are suppose to be fully respected and/or other contacts result as a by-product of the measure [9], except for the case of total closed schools and universities as is happening in Brazil at the time of writing, see more details in the main text. For combination of NPIs, one should take the lowest value in each corresponding row.

| (i) | $g_i$(No NPI) | $g_i$(CSU) | $g_i$(SD60+) | $g_i$(VHQ) | $g_i$(IQ) |
|---|---|---|---|---|---|
| 1 | 1 | 0.25 | 1 | 0.5 | 0.25 |
| 2 | 1 | 0.40 | 1 | 0.5 | 0.25 |
| 3 | 1 | 0.96 | 1 | 0.5 | 0.25 |
| 4 | 1 | 0.96 | 1 | 0.5 | 0.25 |
| 5 | 1 | 0.98 | 1 | 0.5 | 0.25 |
| 6 | 1 | 0.98 | 1 | 0.5 | 0.25 |
| 7 | 1 | 1 | 0.25 | 0.5 | 0.25 |
| 8 | 1 | 1 | 0.25 | 0.5 | 0.25 |
| 9 | 1 | 1 | 0.25 | 0.5 | 0.25 |

*H* means the healthcare demand due to hospitalized cases requiring critical attention in ICU, see Table 3. This equation can be coupled to the previous system (1) to estimate the time evolution of the health system demand (H). It is worth mentioning that the critical cases require long hospitalization, meaning that we will neglect the beds vacancies generated by recoveries or deaths. Although the number of ICUs is massively concentrated in capitals and big urban centers [14], this in-homogeneity is not considered in our model, that should be seen as an average over all different regions in Brazil. The model can be improved to account for a contact matrix $C_{ij}$ given the probability of contact between the age groups, however it would be only a heuristic guess, so we decide not to include it here, see for instance [27, 28]. Although the contact matrix is still relevant, the infectiousness of infected individuals seems to be almost the same for distinct symptomatic outcomes [29]. It is also worth mentioning that SIRD-like models can be comparable with more complex models such as random graphs networks [30–34].

## 1.1 Modelling non-pharmaceutical interventions (NPIs)

The parameter $g_i$ represents the NPI policies and, as in [9], they are not supposed to have full compliance of the individuals. Further, as a by-product such NPIs might generated other kinds of social contacts, for instance, those due to the essential services that continue running even in a mandatory quarantine. For combination of NPIs, one should take the lowest value in each corresponding row of Table 2. So, $g_i$ influences directly the spread of the disease, having strong effect on the efficacy of the infection process and can be interpreted as alterations of the

**Table 3. Mortality and critical hospitalized percentages per age group.**

| Age Range | Mortality[1001] [1, 2, 9] ($\gamma$) | Hospitalised[1001] [1, 9] (c) |
|---|---|---|
| 0-9 | 0.002% | 0.005% |
| 10-19 | 0.006% | 0.015% |
| 20-29 | 0.03% | 0.06% |
| 30-39 | 0.08% | 0.16% |
| 40-49 | 0.15% | 0.31% |
| 50-59 | 0.6% | 1.25% |
| 60-69 | 2.2% | 4.55% |
| 70-79 | 5.1% | 10.5% |
| 80+ | 9.3% | 19.36% |

$\beta_i$ parameter, resulting in an effective $\beta_i^{eff} = g_i\beta_i$ due to the imposed control measure. It comes from the reasonable assumption that in the early stage of the infection $S \approx N$, therefore it fights the infection by reducing the number of susceptible persons. In our approach it reflects the amount of susceptible individuals undergoing the specific control measure and $g_i S_i$ represents the fraction of $S_i$ not complying with the policy $g_i$. Ahead we discussed the NPIs considered in this work along with the expected response from the population to these measures. In all cases, the compliance is not 100% effective [9] since that, for instance, many essential services are needed, so even in an intense and mandatory quarantine we supposed the 25% of the susceptible persons are still well exposed to the infection (see the fraction 0.25 in the last column of Table 2).

In the lack of a vaccine, it is improbable that no NPI ($g_i = 1$) policy would be applied. To estimate $g_i$ (reflecting the fraction of individuals still susceptible after the measure is applied) for the different NPIs and age groups we follow a approach as in Ref [9]. For the case of closure of school and universities (CSU), we use the census-based data of the number of students per age group to reduce the respective number of susceptible individuals. They are all supposed to be uninfected in the early stage of the epidemic and we assume that 100% of this target population will not disobey the policy as all schools and universities are closed in Brazil. The third column of Table 2 is the unit minus the values in the forth column of Table 1. For social distancing of people over 60 years old (SD60+) we assume that 75% will comply with this policy, meaning that 25% will leak the isolation, therefore the fraction 0.25 in the last three rows of Table 2. For voluntary home quarantine (VHQ) and social distancing of the entire population as an intense quarantine (IQ) the compliance is 50% and 75%, meaning that the remaining quantity of susceptible individuals in each age group is factor of 0.5 and 0.25 of their population, respectively. See Table 2 to check $g_i$ for each NPI. As the measures are not capable of producing effects instantaneously, taking in general $t_\beta = 14$ days be effective [22], we have to apply a modulation in the $g_i$ parameter in a similar fashion as done in Ref. [22]. Thus, we model $g_i$ as

$$g_i \rightarrow g_i \frac{1}{1 + \exp{-(t - t_{NPI})/t_\beta}}, \tag{6}$$

where this modulation function is the sigmoid function, $t_{NPI}$ is the date at which the measure is implemented and $t_\beta$ is the number of days it takes to produce effects.

## 1.2 Initial conditions and model calibration

For the model to reflect well the scenarios for the next days according to the adopted public policies, one has to inform the proper initial conditions as well as the model parameters. The corresponding initial conditions ($S_i(0)$, $I_i(0)$, $R_i(0)$, $D_i(0)$) were taken from the data available for the Brazilian case at the time of writing. Differently, from what has been done by many countries, the data from the national ministry of health is not informed in an age-structured manner. For that reason, we collected the data from epidemiological bulletins from the health secretaries from the states of São Paulo, Rio de Janeiro, Ceará, Minas Gerais, Pernambuco, Mato Grosso do Sul, Alagoas and the Distrito Federal, counting 730 infected cases as of 21th March 2020 (the initial time step in our simulations), corresponding to around 65% of the confirmed cases at that date.

We assume that this distribution reflects well the overall national distribution. So the input for $I_i(0)$ is the multiplication of the infected distribution $p[I_i(0)]$ by the official number of infected persons at the March 21, 2020, that is, $N_{infected} = 1128$. The death distribution is more accurate given that the national ministry of health provides this information directly, as informed in the third column of Table 4. Thus, the input $D_i(0)$ is also the multiplication of the

**Table 4. Infected and death percentages per age group in Brazil as of 21th March, 2020.** $S_i(0)$ is just multiplication of corresponding age group population percentage (second column in Table 1) by the total population $N$, discounted the corresponding number of infected in the age group. $R_i(0) = 0$ as at the time of writing it was not a significant data (just 6 recovered individuals). It is a common assumption for early stages of the infection, as is the case in Brazil and in many other countries.

| Age Range | $p[I_i(0)]$ (%) | $p[D_i(0)]$ (%) |
|---|---|---|
| 0-9 | 0 | 0 |
| 10-19 | 0.2 | 0.04 |
| 20-29 | 0.2 | 1.1 |
| 30-39 | 0.2 | 3.4 |
| 40-49 | 0.4 | 4.3 |
| 50-59 | 1.3 | 8.2 |
| 60-69 | 3.6 | 11.8 |
| 70-79 | 8 | 16.6 |
| 80+ | 14.8 | 18.4 |

death distribution $p[D_i(0)]$ by the total number of dead individuals at the time of writing ($N_{dead}$ = 92). The input $S_i(0)$ is just multiplication of corresponding age group population percentage (second column in Table 1) by the total population $N$, discounted the corresponding number of infected in the age group. Because at the time of writing the number of recoveries is negligible (just 6 recovered individuals), we set $R_i(0) = 0$. We remark that this is a common assumption to model infections at early stages, as is the case in Brazil and in many other countries.

The calibration of the model requires robust data so that the model parameters can be as realistic as possible. In the absence of such that for the Brazilian case, we used some data reported in studies with the largest number of individuals with an age-distributed fatality rate ($\gamma_i$) [1] and the percentage of persons undergoing critical intense care ($c_i$) [2]. For the recovery rate ($\alpha_i$) we use the assumption that $\alpha_i = (1 - \gamma_i)^*r$, where $r$ is the overall fraction of recovery in the closed cases, known so far to be $r = 0.82$, meaning that 82% of those who did not succumb to the disease are now healed [7]. This does not imply a overall death ratio of 18%, since it accounts only for closed cases, in which one can compute statistics. Although this fraction could change over time, the statistics is reliable since the number of total closed cases is $N_{closed}$ = 191, 623, roughly 25% of the confirmed cases at the time of writing. The parameter $\beta_i$ describes the efficacy of the infection process and can be measure indirectly. Our first assumption is that this efficacy depends weakly on the age group, therefore $\beta_i = \beta$ is a constant vector. The effective value of $\beta$ can be computed as $R_0 = \frac{\beta}{\bar{\alpha_i + \gamma_i}}$, where $\bar{x}$ stands for the mean value of the variable $x$ and $R_0$ reproduction number already defined and calculated to be in the range 1.5–6.0 in many countries [9, 11]. We will estimate the $R_0$ by performing a fit in our model with current data available. Given the current control measures in Brazil (a combination of CSU, HVQ and SD60+) our current $g_i$ is the concatenation of the corresponding columns in Table 2, taking the lowest values. This was in accordance with the overall estimates via Google's Community Mobility tool around April 1 for Brazil [38].

The data fitting shown in Fig 2 indicates that in Brazil the $R_0 \approx 3$ is still high nevertheless the current control measures. It means that one infected person, on average, is contaminating 3 other individuals. A clear indication that Brazil should not relax the NPIs adopted so far. In fact, as we will show in the next section, even stronger measures are going to be needed to avoid surpassing the capacity of the national health system and, consequently, minimize the number of severe cases and deaths. As we seek to estimate the demand for ICUs and it

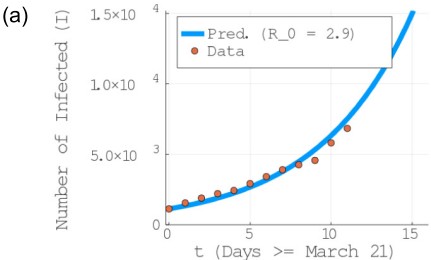

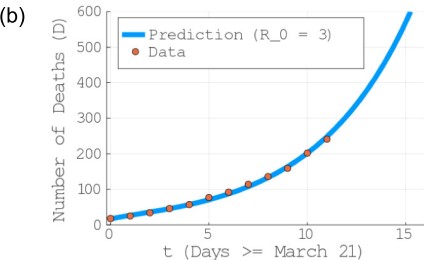

**Fig 2. Estimation of the parameter $R_0$.** Data fitting for: (top) the evolution of the number of deaths (D) and (bottom) number of infected persons (I).

depends heavily on the number of infected (see Eq 6), we are going to use to value of $R_0$ which best fits the variable $I$, so $R_0 = 2.9$ even though the fit for $D$ seems better. It implies $\beta = R_0 * (\bar{\alpha}_i + \bar{\gamma}_i) = 2.39$.

## 2 Results and discussion

With the model initialized and callibrated as described in the previous section, we obtain the results by solving the system of ordinary differential equations (ODEs) given by Eqs (1) to (5) with the Julia Programming Language's package *DifferentialEquations* [39].

Our main result is to show that, even though the current NPI measure taken in Brazil have led to a substantial decrease in the number of infections as compared to no NPI since the beginning of the reported cases in Brazil (see red and blue curves in Fig 3 (bottom)), the current measures are not enough to prevent the collapse of the health system in a short period of time with million of infected persons (see Figs 3 (top) and 4).

Even with the combination of CSU, HVQ and SD60+ taking place, our model predicts millions of infected individuals, with a peak taking place around the middle of May, 2020 (see Fig 3 (top)) in agreement with the projection for Brazil in Ref. [40], and consequently an exponential increase in the demand for ICUs. In fact, as can be seen in Fig 4, already at the end of April we will surpass the current capacity of ICUs. The health system is still under treat in the current scenario, indicated to collapse by the end of Abril, 2020 (around 30 days after t0 == March 21, 2020), vigorously crossing the 5500 ICUs barrier. Moreover, the scenario is even worse if the imposed NPIs are relaxed (as being constantly debated by the Brazilian federal government), pointing out tens of millions of infected individuals, see black curve in Fig 3 (bottom).

On the positive side, we have also identified a window of 25 days—from the March 21st to April 16th—in which, similarly to what has been done in China, if more severe control measures are be applied, one can control the virus spread and keep the ICU demand below the threshold (see the green curve in Fig 4). Given that the actual NPI scenario in Brazil is

(a)

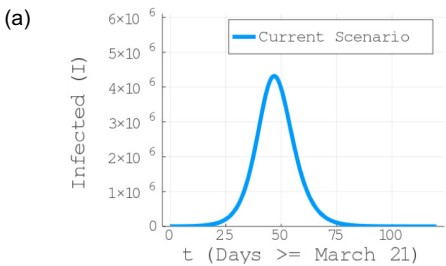

(b)

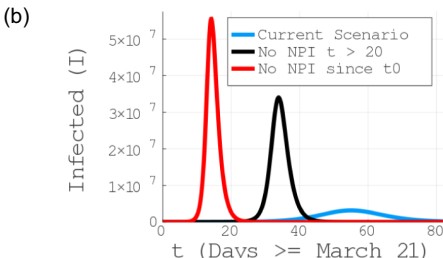

**Fig 3. Infected individuals as a function of time for the current NPI measures being taken in Brazil.** The current measures not only have reduced the number of infections but also moved the peak of the contamination to later date, given the authorities the necessary time for preparation. The model has a prognostics of million of infected persons in the current scenario (with a peak around the end of the first half of May, 2020) and tens of millions in the case of returning to a zero NPI scenario, if adopted 20 days after t0 (March, 21), meaning $t_{NPI}$ == April, 11. Notice a current relatively slow progress, until 30 days from March, 21 (April, 21). Stricter control measures, such as an intense quarantine, are more effective within this window.

represented by the combination of the lowest values of the $g_i$ functions in Table 2 for CSU, SD60+ and VHQ, the only possible measure is an intense quarantine (last column of Table 2). Except a intense quarantine, all the other scenarios are more catastrophic, meaning a faster collapse of the system. Although an emergence expansion and/or other measures can be tried, it is unlikely to keep the pace with the exponential spread of the infection. Even if we are underestimating the number of available ICUs, the saturation will still take place for the current scenario (as it shows a positive concavity), being only reversed with the intense quarantine implementation.

In our model, this intense quarantine is supposed to be applied around 20 days after our initial time (March, 21). As discussed in the NPI section, no measure is instantaneous efficient, taking on average a time $t_\beta \approx 14$ days to be completely noticed [22]. In spite of that, notice that

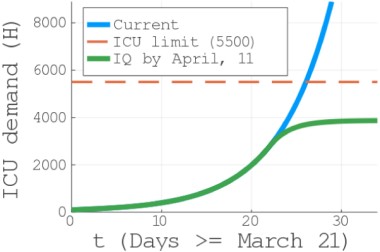

**Fig 4. ICU beds demands for distinct scenarios.** ICU beds demand for the current scenario of closed schools and universities and voluntary home quarantine (blue line). Note that the health system is still under treat in the current scenario, indicated to collapse by April 21, 2020 (30 days after t0 == March 21, 2020). In green, the ICU demand if an intense quarantine is imposed by April, 11 (20 days after t0 == March 21, 2020).

**Table 5. Estimated number of infected $I_i$ for the actual control measures and considering changes to one of the following scenarios (starting at April 11th): (No NPI), closure of schools and universities (CSU), social distancing of people with more than 60 years old (SD60+), voluntary home quarantine (VHQ) and intense quarantine(IQ).**

| Age | Cur. | (No NPI) | (CSU) | (SD60+) | (VHQ) | (IQ) |
|---|---|---|---|---|---|---|
| 0-9 | 280K | 4.70M | 1.35M | 4.08M | 1.3M | 2900 |
| 10-19 | 460K | 5.11M | 2.13M | 4.44M | 1.41M | 5000 |
| 20-29 | 600K | 5.48M | 4M | 4.76M | 1.51M | 6700 |
| 30-39 | 610K | 5.55M | 4.06M | 4.82M | 1.53M | 6700 |
| 40-49 | 510K | 4.67M | 3.45M | 4.05M | 1.29M | 5700 |
| 50-59 | 420K | 3.85M | 2.84M | 3.34M | 1.06M | 4700 |
| 60-69 | 150K | 2.58M | 1.92M | 869K | 712K | 1600 |
| 70-79 | 81K | 1.35M | 1M | 455K | 373K | 800 |
| 80+ | 44K | 738K | 550K | 248K | 203K | 450 |
| **Total** | **3.15M** | **30.47M** | **21.2M** | **26.96M** | **9.4M** | **34.5K** |

the increase in the ICU demand is rapidly contained. Moreover, it is important to mention that such a intense quarantine should last enough time until we reach a plateau in the ICU demand. Not only the quarantine protects the health system, but also corresponds to a minimization of the number of deaths, massively reducing it.

In Tables 5 and 6, we show the estimated number of infected people and deaths, respectively, in the current scenario as well as if other NPI measures start to take place at April 11th (20 days after March 21st). Observe how the total of infected is still very high in the current scenario: around 3.15 million infected individual, with an astonishing 393 thousands deaths. These estimations is based from the initial date up to 150 days ahead, where the disease is modelled to be controlled. As discussed, the only scenario which considerably reduces the number of infected persons and deaths is an intense quarantine. It reduces to 34.5 thousand infections and 1300 deaths. In particular, we notice that changing the current NPI to SD60+ (f social distancing of people above 60 years old) as been currently discussed by the Brazilian federal government, is completely catastrophic. As this group corresponds to the majority of critical care demands and deaths, their isolation is certainly necessary. However, taken alone it leads to ≈27 millions infected and 723 thousand dead individuals (see the fifth columns in Tables 5 and 6). An important observation is that our predictions are likely to be a lower bound to the actual numbers, since the confirmed cases are potentially underestimated given the lack of a

**Table 6. Estimated number of deaths $D_i$ for the actual control measures and considering changes to one of the following scenarios (starting at April 11th): (No NPI), closure of schools and universities (CSU), social distancing of people with more than 60 years old (SD60+), voluntary home quarantine (VHQ) and intense quarantine(IQ).**

| Age | Cur. | (No NPI) | (CSU) | (SD60+) | (VHQ) | (IQ) |
|---|---|---|---|---|---|---|
| 0-9 | 140 | 590 | 230 | 600 | 350 | 0 |
| 10-19 | 700 | 1930 | 1090 | 1900 | 1200 | 2 |
| 20-29 | 4.5K | 10.4K | 9.8K | 10K | 6500 | 15 |
| 30-39 | 12K | 28K | 26.3K | 27K | 17.5K | 40 |
| 40-49 | 19K | 44K | 42K | 43K | 27.5K | 65 |
| 50-59 | 63K | 145K | 138K | 142K | 91K | 210 |
| 60-69 | 86K | 357K | 342K | 146K | 223K | 280 |
| 70-79 | 104K | 433K | 414K | 176K | 270K | 350 |
| 80+ | 44.8K | 738K | 550K | 248K | 203K | 450 |
| **Total** | **393K** | **1.45M** | **1.38M** | **723K** | **905K** | **1300** |

widespread clinical testing in Brazil with more accurate data, it is possible that the fitting in Fig 2 would produce an even higher value for $R_0$. In addition, our model can be applied to other countries or regions with minimal adaptation, since it will require only updates in Tables 1–3.

Finally, although we have employed many data-driven assumptions, the results presented here may still underestimate the threat to the national health system due to the particular social problems in Brazil, such as: (i) high level of cardiopathologies, a reported relevant comorbidity; (ii) considerable number of people without proper water and wasting supplies; (iii) large number of people living in the same house in peripheral zones due to housing deficit, (iv) high density of obesity cases and some other immune suppressant diseases, just to cite a few. On the positive side, there can be a potential defense against SARS-Cov-2 for those BCG-vaccinated [41], which belongs to the universal vaccination program in Brazil [42]. Furthermore, the public health system in Brazil has great capillarity, in principle being capable to identify potential cases in the very beginning of the symptoms. However, that also requires widespread clinical tests to identify infected individuals.

## Conclusion

In this work we proposed a data-driven age-structured census-based SIRD-like epidemiological model capable of forecasting the spread of COVID-19 in Brazil in a number of NPI scenarios. We remark that our approach is fairly general and thus can be applied to treat particular regions or cities, if the required data is available.

As we have shown, the early NPI measures taken by states and cities such as the total closure of schools, universities and non-essential services, the social distancing and isolation of individuals above 60 years and the voluntary home quarantine have already lead to significant reduction in the number of infections as well as delaying the time for the peak of contamination. Thus, these measures have been extremely important to give the authorities the necessary time for the adapting and preparing before the peak of the epidemic.

Notwithstanding, the current measures are not enough. Our model predicts that even if the current NPIs are not relaxed, as early as mid April the number of severe cases requiring hospitalization will surpass the current number of available ICUs, starting the collapse of the health system. However, a intense quarantine, if implemented in the following days, can rapidly change the increase in the number of infections and keep the demand for ICUs below the threshold, amounting to hundred of thousands of saved lives. On the other hand, we demonstrate that the relaxation of the already imposed control measures in the next days, as currently debated at some sphere of the Brazilian federal government, would be catastrophic, with a total death toll passing the one million mark.

In a nutshell, a continued quarantine, tighter than the current one and with a duration of a couple of weeks, is most likely the only solution to avoid the collapse of the health systems in Brazil. We hope that the gigantic difference in numbers, showing how different measures can lead to a reduction in infections and deaths of the order of hundreds of thousands or even millions, can provide a rational guide for the future decisions by the competent authorities.

## Acknowledgments

We thank the Brazilian agencies MCTIC and MEC. AC also acknowledges UFAL for a paid license for scientific cooperation at UFRN.

## Author Contributions

**Conceptualization:** Askery Canabarro, Elayne Tenório, Samuraí Brito, Rafael Chaves.

**Data curation:** Askery Canabarro, Samuraí Brito, Rafael Chaves.

**Investigation:** Askery Canabarro, Elayne Tenório, Renato Martins, Laís Martins, Samuraí Brito, Rafael Chaves.

**Methodology:** Askery Canabarro, Elayne Tenório, Renato Martins, Laís Martins, Samuraí Brito, Rafael Chaves.

**Writing – original draft:** Askery Canabarro, Elayne Tenório, Samuraí Brito, Rafael Chaves.

**Writing – review & editing:** Askery Canabarro, Elayne Tenório, Samuraí Brito, Rafael Chaves.

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
