## [Decision Letter · Decision Letter 0]

28 May 2020

PONE-D-20-09888

Data-Driven Study of the COVID-19 Pandemic via Age-Structured Modelling and Prediction of the Health System Failure in Brazil amid Diverse Intervention Strategies

PLOS ONE

Dear Dr. Canabarro,

Thank you for submitting your manuscript to PLOS ONE. After careful consideration, we feel that it has merit but does not fully meet PLOS ONE’s publication criteria as it currently stands. Therefore, we invite you to submit a revised version of the manuscript that addresses the points raised during the review process. Mainly by one of the referees who decided to reject the paper

We look forward to receiving your revised manuscript.

Kind regards,

Lidia Adriana Braunstein, Phd in Physics

Academic Editor

PLOS ONE

Journal Requirements:

We acknowledge the John Templeton Foundation via the Grant Q-CAUSAL No. 61084,

the Serrapilheira Institute (Grant No. Serra-1708-15763), the Brazilian National Council

for Scienti c and Technological Development (CNPq) via the National Institute for Science

and Technology on Quantum Information (INCT-IQ) and Grants No. 307172/2017-1 and

No. 406574/2018-9 and No 423713/2016-7, the Brazilian agencies MCTIC and MEC. AC

also acknowledges UFAL for a paid license for scienti c cooperation at UFRN .

4. We note you have included a table to which you do not refer in the text of your manuscript. Please ensure that you refer to Table IV in your text; if accepted, production will need this reference to link the reader to the Table.

Reviewers' comments:

Reviewer's Responses to Questions

**Comments to the Author**

1. Is the manuscript technically sound, and do the data support the conclusions?

Reviewer #1: No

Reviewer #2: Yes

Reviewer #3: Yes

2. Has the statistical analysis been performed appropriately and rigorously? 

Reviewer #1: N/A

Reviewer #2: Yes

Reviewer #3: Yes

3. Have the authors made all data underlying the findings in their manuscript fully available?

Reviewer #1: Yes

Reviewer #2: Yes

Reviewer #3: Yes

4. Is the manuscript presented in an intelligible fashion and written in standard English?

Reviewer #1: Yes

Reviewer #2: Yes

Reviewer #3: Yes

5. Review Comments to the Author

Reviewer #1: The work "Data-Driven Study of the COVID-19 Pandemic via Age-Structured

Modelling and Prediction of the Health System Failure in Brazil amid Diverse

Intervention Strategies" presents a model built by coupling SIR models for

different ages, thus introducing an age structure.

The true variables of the model are fractions of populations (s=S/N, r=R/N,...)

where N is the population of Brazil, more than 200 million people. The

age-populations are coupled by a matrix of contagious contacts which originally

is a multiple of matrix with all ones, 1, in its entries, but later is modified

to accommodate what the authors speculate is the translation into parameters of

the non pharmaceutical interventions (NPI).

There is no support in the biomedical literature for the original (before

interventions) contact matrix, actually mild cases are expected to be less

contagious than severe cases (at e very primitive level, think that a good

number of mild cases do not present cough, one of the predominant mechanisms

for contagious). The immunological system is also expected to wear off with age in

average. Thus, the contagious contact matrix the authors use as a first step is

not supported by present knowledge of SARS-CoV-2. The same can be said for

recovery times.

The authors leave aside that the SIR model does not represent a proper

progression of contagiousness.

Furthermore, the homogeneous mixing implicitly assumed of the population is known

to be a problematic assumption to results in exaggerated number of contagious.

In turn, the reduction factors associated to the NPI are rather arbitrary and

more likely than over-optimistic. The effectiveness of a quarantine is related

to the social structure and is going to be rather heterogeneous. It is almost

impossible to prevent contact to families living together and is almost

impossible to restrain to their home to those which leave under precarious

conditions. I am thinking of shanty towns (Favelas), Com-unitary isolation is

more likely to happen in such social conditions. As a consequence, results such

as those in Figure 3 are truly fabulated.

I fail to see a reason to recommend to PLOS' readership the present manuscript.

Reviewer #2: In this work the authors propose a data-driven age-structured census-based SIRD-like epidemiological model capable of forecasting the spread of COVID-19 in Brazil. They model the current scenario of closed schools and universities, social distancing of individuals above sixty years old and voluntary home quarantine, and show that it led to a considerable reduction in the number of infections as compared with a scenario without any control measures. However, the authors predicts that the current measures are not enough to avoid overloading the health system, since the demand for intensive care units will soon surpass the number available and that an urgent intense quarantine might be the only solution to avoid this scenario and, consequently, minimize the number of severe cases and deaths.

The paper is well written and clear and I recommend that the manuscript be published. However, I have a few comments that I believe may improve the ms.

1. In page 4, after the reproduction number R_0 is definited, the same magnitud is named as R_n . At least I have missing something I think the authors are talking about the same magnitud.

2. In page 7, line 3 from the bottom, the phrase “The third column of Table II is the unit minus the values in the third column of Table I” it is wrong. I think that the third column of Table II is the unit minus the values in the fourth column of Table I. Please check this.

3. Coronavirus report given by the World Health Organization (WHO) indicates that the total number of infected in Brasil by May 1st is of 78162 confirmed cases, less than 10^5 (“https://www.who.int/emergencies/diseases/novel-coronavirus-2019/situation-reports”). This value is very different from the one estimated by the model (showed in Fig. 3a), which is over the 10^6 cases. Can the authors explain this huge difference?

Reviewer #3: The manuscript is technically solid. The data used in the analysis is available at Johns Hopkins University COVID-19 database. The method used is well known and has been used in similar articles. The article has been published in a preprint version, including the legend that has not been peer-reviewed.

I suggest that the authors should take into account the contact matrix in the analysis, because the present analysis shows the direct effect of the NPI interventions. If the contact matrix are incorporated the authors can include the indirect effects generated from contacts between different age groups.

The title of Table IV should be reviewed, since it should mention the case fatality rate, instead of mortality.

Table V should specify whether the information on those infected is until the entire population finishes infecting or until a certain date.

In the results, it would be of interest for health system planning policies that the demand for UTI beds be presented in each scenario.

The conclusions are well developed and conform to those presented in the introduction.

6. PLOS authors have the option to publish the peer review history of their article (what does this mean?). If published, this will include your full peer review and any attached files.

Reviewer #1: No

Reviewer #2: No

Reviewer #3: No

---

## [Author Response · Author response to Decision Letter 0]

16 Jun 2020

We have uploaded a complete response letter to all reviewers' questions.

---

## [Decision Letter · Decision Letter 1]

7 Jul 2020

Data-Driven Study of the COVID-19 Pandemic via Age-Structured Modelling and Prediction of the Health System Failure in Brazil amid Diverse Intervention Strategies

PONE-D-20-09888R1

Dear Dr. Canabarro,

We’re pleased to inform you that your manuscript has been judged scientifically suitable for publication and will be formally accepted for publication once it meets all outstanding technical requirements.

Kind regards,

Lidia Adriana Braunstein, Phd in Physics

Academic Editor

PLOS ONE

Additional Editor Comments (optional):

Reviewers' comments:

Reviewer's Responses to Questions

**Comments to the Author**

1. If the authors have adequately addressed your comments raised in a previous round of review and you feel that this manuscript is now acceptable for publication, you may indicate that here to bypass the “Comments to the Author” section, enter your conflict of interest statement in the “Confidential to Editor” section, and submit your "Accept" recommendation.

Reviewer #3: All comments have been addressed

2. Is the manuscript technically sound, and do the data support the conclusions?

Reviewer #3: Yes

3. Has the statistical analysis been performed appropriately and rigorously? 

Reviewer #3: Yes

4. Have the authors made all data underlying the findings in their manuscript fully available?

Reviewer #3: Yes

5. Is the manuscript presented in an intelligible fashion and written in standard English?

Reviewer #3: Yes

6. Review Comments to the Author

Reviewer #3: The authors have adequately addressed my comments raised in the previous round of review and I feel that this manuscript is now acceptable for publication

7. PLOS authors have the option to publish the peer review history of their article (what does this mean?). If published, this will include your full peer review and any attached files.

Reviewer #3: **Yes: **Laura Soledad Lamfre

---

## [Editor Report · Acceptance letter]

22 Jul 2020

PONE-D-20-09888R1 

Data-Driven Study of the COVID-19 Pandemic via Age-Structured Modelling and Prediction of the Health System Failure in Brazil amid Diverse Intervention Strategies 

Dear Dr. Canabarro:

I'm pleased to inform you that your manuscript has been deemed suitable for publication in PLOS ONE. Congratulations! Your manuscript is now with our production department. 

Kind regards, 

on behalf of

Dr. Lidia Adriana Braunstein 

Academic Editor

PLOS ONE